# The Posttranscriptional Mechanism in *Salvia miltiorrhiza* Bunge Leaves in Response to Drought Stress Using Phosphoproteomics

Jin Zhang [1,2,†], Jingyu Li [1,†], Yuekai Su [1], Zhenqiao Song [1] and Jianhua Wang [1,*]

1  State Key Laboratory of Crop Biology, College of Agronomy, Shandong Agricultural University, Tai'an 271000, China; zhjin0214@163.com (J.Z.); sdaulj@163.com (J.L.); suyuekai163@163.com (Y.S.); szqsdau@163.com (Z.S.)
2  Taishan Academy of Forestry Sciences, Tai'an 271000, China
*  Correspondence: sdauwangjh@163.com
†  These authors contributed equally to this work.

**Abstract:** Drought stress is a major constraint to the quality and production of *Salvia miltiorrhiza* Bunge (Danshen). This study aimed to investigate the posttranslational molecular mechanisms in *S. miltiorrhiza* leaves in response to drought stress using quantitative phosphoproteomics analysis. *S. miltiorrhiza* plants were stressed by withholding water for two (moderate drought stress) and four weeks (high drought stress). Leaf samples were prepared with tandem mass tag labeling. Liquid chromatography-tandem mass spectrometry was performed for the quantitative phosphoproteomics. Bioinformatics methods were used to identify the phosphosites and phosphoproteins that had significantly changed phosphorylation levels upon drought stresses. A total of 119 common phosphoproteins were significantly changed by both high and moderate drought stresses. The phosphorylation levels of proteins related to protein processing, photosynthesis, RNA binding, and splicing were significantly changed upon high drought, not moderate drought. Additionally, we identified that the Ser phosphorylation levels of most proteins related to terpene metabolism and RNA splicing were regulated by drought stresses. The Ser and Thr phosphorylation levels of energy metabolism proteins (including FBA2/8, PPC4, and PPCC) and heat shock proteins (including HSP70 and HSP90) were upregulated by drought stresses. Our study showed the posttranscriptional mechanisms in *S. miltiorrhiza* leaves in response to drought stress.

**Keywords:** post-translational phosphorylation; phosphoproteomics; drought stress; *Salvia miltiorrhiza*

## 1. Introduction

With the changing global climate, the occurrence of natural disasters that adversely affect crop production is becoming more frequent and intensive in many parts of the world. Osmotic stress caused by drought is a major constraint to crop yield. Drought is abiotic stress that has a devastating effect on crop production and quality [1,2]. In the face of drought conditions, plants have evolved a series of molecular and physiological responses and adaptation mechanisms to tolerate or reduce damages caused by this stressful condition [1,3]. The investigation of the underlying molecular mechanism in the plant in response to drought stress is of significance for improving crop yield.

Research in plant genomics and proteomics have been widely used to identify the underlying molecular mechanisms in response to abiotic stresses [4–7]. Some of the genes and proteins responsible for the increased tolerance and adaptation to drought stress have been identified [1,7,8]. The earliest stress-responsive changes in gene expression cannot accurately reflect the protein profiles in response to the same conditions because of the post-translational modifications of proteins [4,9]. Post-translational modifications, including monoubiquitination, glycosylation, and phosphorylation affect protein localization, stability, activity, and specific cellular responses [9–11]. Accordingly, the drought-stress responses in the plant could be controlled by post-translational modifications of protein.

Protein phosphorylation is a major and reversible post-translational modification that regulates almost all aspects of protein localization, processing, folding, function, stability, and activity. Recent research in plant phosphoproteomics have largely focused on the identification and characterization of phosphorylated proteins in response to stressful conditions [6,12–14]. The changed phosphorylation levels of proteins in response to stress conditions suggested that the post-translational phosphorylation is responsible for the increased adaption and tolerance to abiotic stresses. Recently, the large-scale phosphoproteomics analysis had been performed in model plants including *Arabidopsis thaliana* [15], rice [16], maize [6], and wheat [17], and non-model plants including *Brachypodium distachyon* [18], *Ammopiptanthus mongolicus* [12], and pepper [19]. However, there is far less study of phosphoproteomics focusing on *Salvia miltiorrhiza*.

*S. miltiorrhiza* Bunge (Danshen in Chinese) is one of the most important and popular traditional Chinese medicine (TCM). It is used as a golden herbal medicine for the treatment of cardiovascular diseases including coronary heart disease, atherosclerosis, myocardial infarction, and ischemia [20,21]. Its water-soluble phenolic acids are of significant bioactivities including anti-inflammatory, antioxidant, and health-promoting activities [21]. Due to its great and significant medicinal values, the social and clinical demand for *S. miltiorrhiza* is increasing gradually. Several studies have shown that drought stress affects the growth and accumulation of active constituents in *S. miltiorrhiza* [22]. However, little is known about the underlying phosphoproteomics mechanism in *S. miltiorrhiza* upon drought stress.

In this study, a quantitative phosphoproteomics analysis was applied to reveal the response of *S. miltiorrhiza* leaves to drought stress. The liquid chromotography–tandem mass spectrometry (LC-MS/MS) analysis identified a large number of phosphorylation sites and proteins that were involved in the responsive mechanism to drought stress in *S. miltiorrhiza* leaves. The related comprehensive mechanism and pathways of phosphoproteins that had significantly changed phosphorylation level upon drought stress would be fully discussed.

## 2. Results

### 2.1. Morphology Change in S. miltiorrhiza Leaves in Response to Drought Stress

The significant changes in the morphology of *S. miltiorrhiza* plants in response to drought stress are presented in Figure 1. After 2-weeks (MD) and 4-weeks (HD) withholding of water, *S. miltiorrhiza* plants were slightly and severely wilting, respectively. The older leaves in the MD group were beginning to shrivel at the end of treatment. In the HD group, *S. miltiorrhiza* leaves were almost dried up, and the youngest leaves were beginning to shrivel.

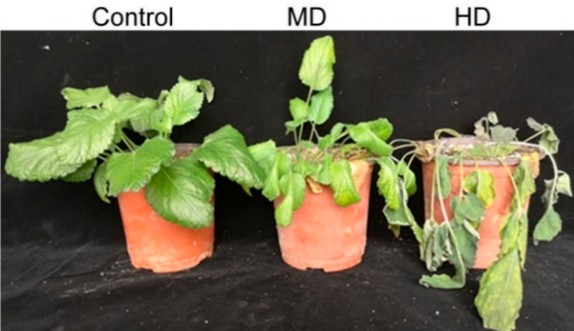

**Figure 1.** The changes in the morphology of *Salvia miltiorrhiza* in response to drought stress. MD and HD indicate moderate drought (2-week withholding water) and high drought stress (4-week withholding water), respectively. CK, control.

### 2.2. Phosphorylation Motifs in the Phosphopeptides

The changes in the phosphoproteins in the youngest leaves were detected using LC-MS/MS and MaxQuant analyses, which resulted in the identification of 9309 phosphosites in 3341 proteins, among which 7354 phosphosites in 2904 proteins contained quantitative

information. The prediction of motif specificity of these 7354 phosphosites showed that there were 69 phosphorylation-based motifs with serine (S) residues, threonine (T) residues, and tyrosine (Y) residues. Among the 69 motifs, nine had motif scores higher than 32. Six motifs including [pSDDD], [pSPxR], [pSPK], [pSPxR], [pSxG], and [pSxD] are presented in Figure 2A. The frequency of the amino acids flanking the phosphorylation sites (Ser and Thr) was evaluated (Figure 2B,C). The P amino acid at the +1 position and the R amino acid at the +2 position had the highest frequency of phosphorylation for both sites. The high frequencies of D (aspartic acid), E (glutamine), K (lysine), P (proline), R (arginine), and S positions were observed in the flanking sequence (−6~+6) of Ser phosphorylation site (Figure 2B).

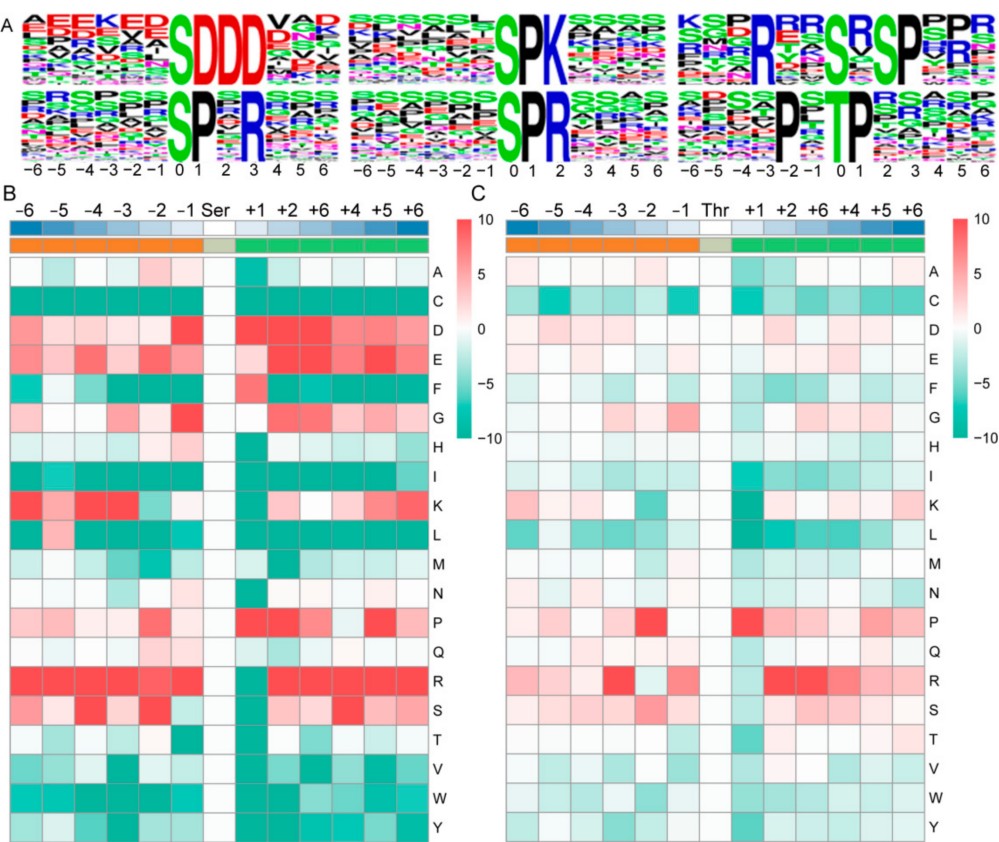

**Figure 2.** Analysis of sequence motifs related to phosphorylation sites. (**A**), the sequence of several motifs of phosphorylation sites. (**B**,**C**) the heatmap for the distribution of amino acids in the flanking sequences of serine and threonine phosphorylation sites, respectively.

### 2.3. Identification of Significantly Changed Phosphopeptides and Phosphoproteins in S. miltiorrhiza Leaves upon Drought Stresses

After filtering phosphorylation sites (phosphosites) with localization probability of lower than 0.75 and data normalization, a total of 7440 phosphosites in 3089 proteins were retained, among which 2209 (71.51%), 425 (13.76%), 197 (6.38%), 138 (4.47%), and 58 (1.88%) proteins were phosphorylated at the Ser, Ser&Thr, Thr, Ser&Tyr, and Ser&Thr&Tyr residues, respectively (Figure 3A). According to the thresholds of |fold change| >1.5 and $p < 0.05$, a total of 859 and 192 phosphosites and 576 and 149 proteins had significantly changed phosphorylation levels in response to high drought (HD) and moderate drought (MD) stress, respectively (Figure 3B). A Venn diagram showed that there were 604 proteins, including 119 proteins common to HD and MD stresses (Figure 3C). The distinct expression profiles of 604 phosphoproteins are presented in Figure S2. Subcellular localization analysis showed that these proteins were mainly located in the nucleus (34.8% and 34.1%, respectively), chloroplast (29.1% and 31.07%), and cytoplasm (17.2% and 18.0%; Figure 3D).

Protein domain analysis showed that the phosphoproteins had a significant increase in phosphorylation level upon HD stress and had PB1 domain, C2 domain, EF-hand domain, and major facilitator superfamily domain (Figure 3E). Decreased phosphoproteins had P-loop and AIG1-type guanine nucleotide-binding domains. In comparison with MD stress, the phosphoproteins that had significantly decreased phosphorylation level upon HD stress possessed heat shock protein 70 kDa (HSP70) C-terminal and peptide-binding domains. These results showed that phosphoproteins with significantly decreased and increased phosphorylation had different functions in *S. miltiorrhiza* leaves upon drought stress.

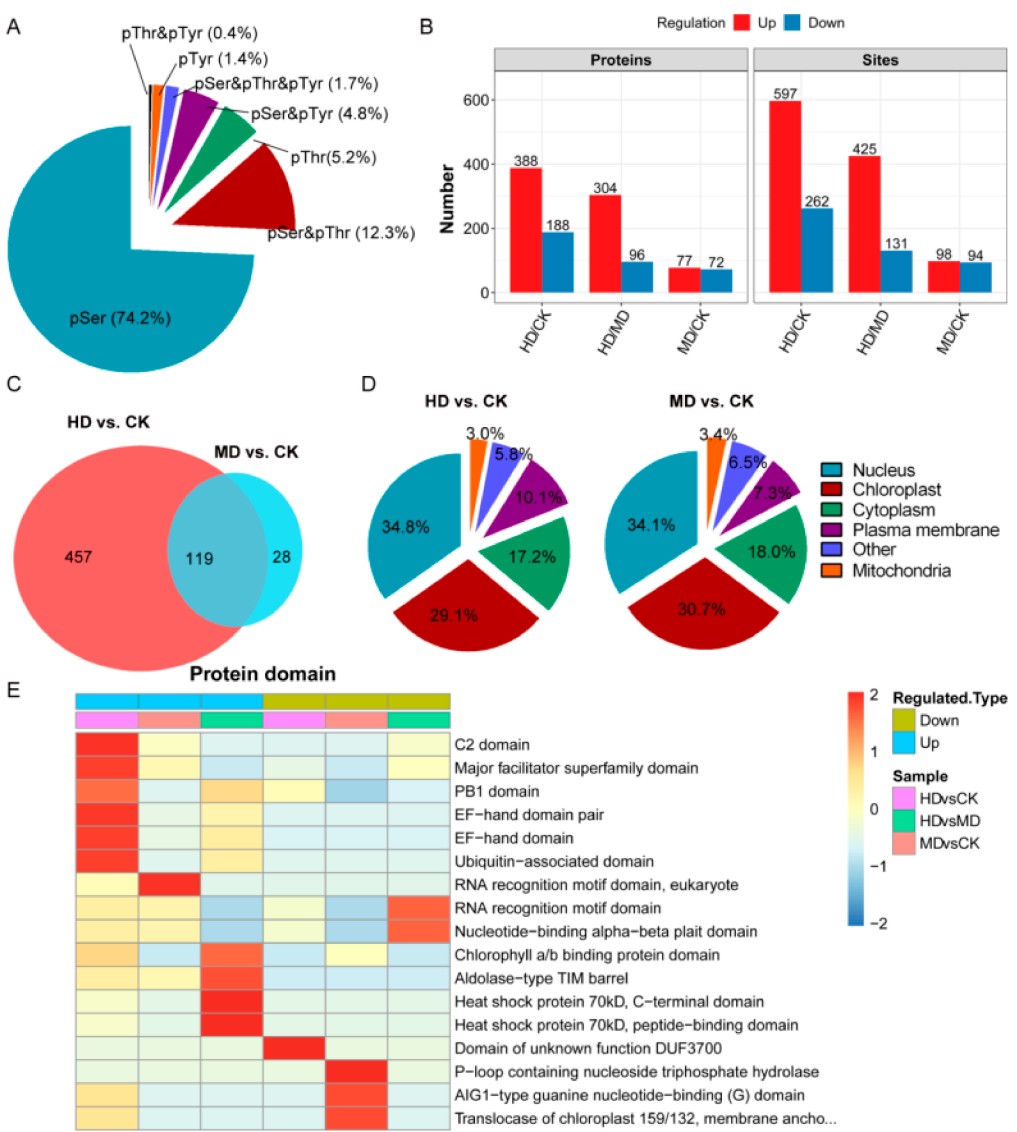

**Figure 3.** The statistics of phosphosites and phosphoproteins in *Salvia miltiorrhiza* leaves. (**A**), the pie chart of phosphosites in *Salvia miltiorrhiza* leaves. (**B**), the histogram of phosphosites and phosphoproteins had significant changes (|fold change| > 1.5 and *p* < 0.05) in the phosphorylation level under high drought (HD) and moderate drought (MD) stress, respectively. (**C**), the Venn diagram of significantly changed phosphoproteins upon HD and MD stress. (**D**), the cellular location analysis of significantly changed phosphoproteins. (**E**), the heatmap clustering of the protein domain derived from significantly changed phosphoproteins upon HD and MD stress, respectively. CK, control.

### 2.4. Functional Classification of Significantly Changed Phosphoproteins in S. miltiorrhiza Leaves upon Drought Stresses

Functional enrichment analysis showed that the significantly changed phosphoproteins upon HD stress were mainly concentrated in KEGG pathways including "carbon

fixation in photosynthetic organisms" (sly00710) and "spliceosome" (sly03040), while MD stress-changed phosphoproteins were associated with "photosynthesis" (sly00195; Figure 4A). Phosphoproteins with increased phosphorylation upon HD and MD stresses were enriched in sly00710, "protein processing in endoplasmic reticulum" (sly04141), and "photosynthesis-antenna proteins" (sly00196; Figure 4B). The phosphoproteins with significantly decreased phosphorylation upon HD and MD stress were associated with sly00195 and sly03040, respectively. The MD stress significantly decreased the phosphorylation levels of proteins related to sly00195 and sly01100 but upregulated the phosphorylation levels of proteins related to sly03040. The HD stress upregulated the phosphorylation levels of proteins related to sly00710, sly04141, sly00196, and "Carbon metabolism" compared with MD stress (sly01200; Figure 4B).

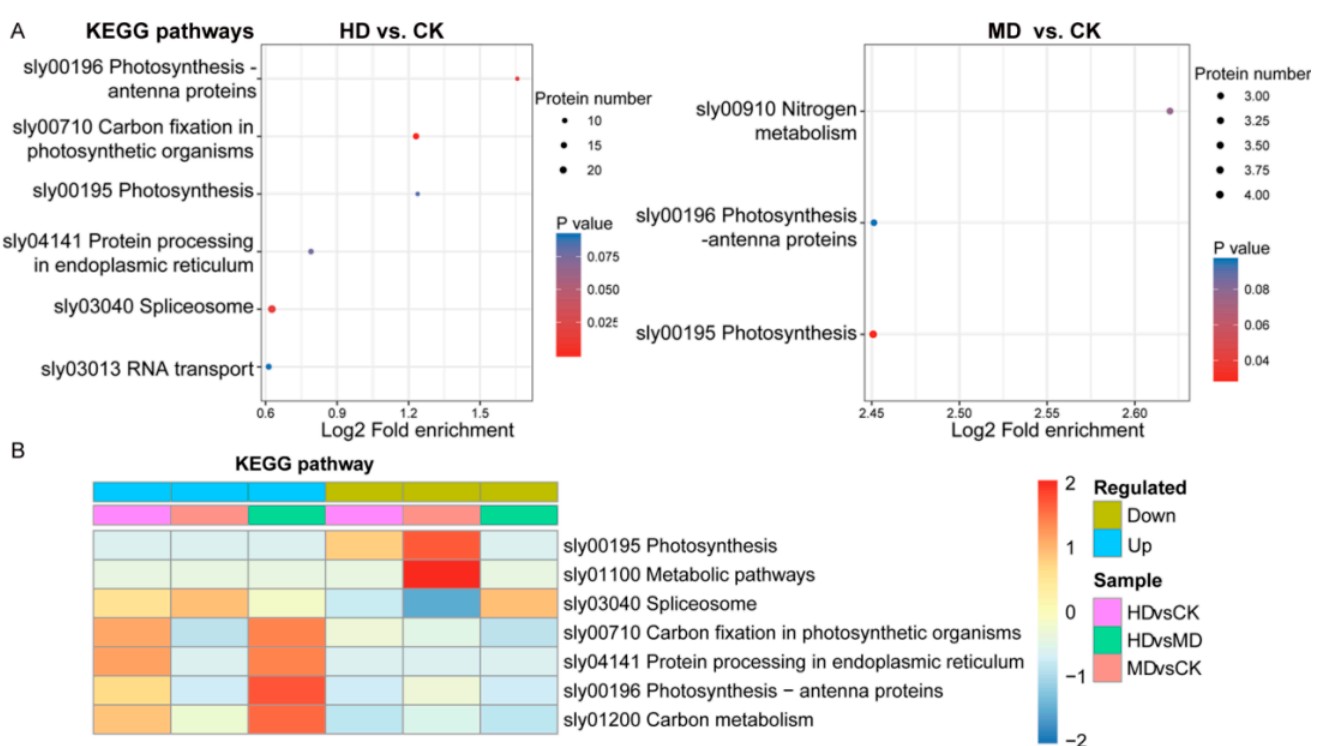

**Figure 4.** The KEGG pathways related to significantly changed phosphoproteins upon drought stress. (**A**), the bubble plot of KEGG pathways that relate to phosphoproteins had significant changes in the phosphorylation level in response to high drought (HD) and moderate drought (MD) stress, respectively. (**B**), the heatmap clustering of KEGG pathways related to phosphoproteins whose phosphorylation levels were significantly decreased and increased by HD and MD stress, respectively. CK, control. KEGG, Kyoto Encyclopedia of Genes and Genomes.

*2.5. Protein–Protein Interaction (PPI) Network of Phosphoproteins in S. miltiorrhiza Leaves upon Drought Stress*

The PPI network derived from significantly phosphorylated proteins upon HD stress consisted of 126 proteins which were mainly clustered into six subsets (Figure 5). The largest subset consisted of 24 proteins, including serine/arginine-rich splicing factors (SR30, SR40, SR34A, and RSZ22) and DExH-box ATP-dependent RNA helicases (RH40, RH56, and RH3). These proteins were related to nucleic acid binding, RNA processing, "RNA degradation" (sly03018), and sly03040. Another subset consisted of one significantly decreased phosphoprotein and 16 significantly increased phosphoproteins, including several heat shock proteins (HSPs), including HSP90B, HSP70, HSP70-HSP90 organizing protein 3 (HOP3), HSFB2B that were centered on HSP70, a probable mediator of RNA polymerase II transcription subunit 37e (MED37E) and other chaperones. These proteins were related to response to stress/stimulus, protein folding, sly04141, and sly03040.

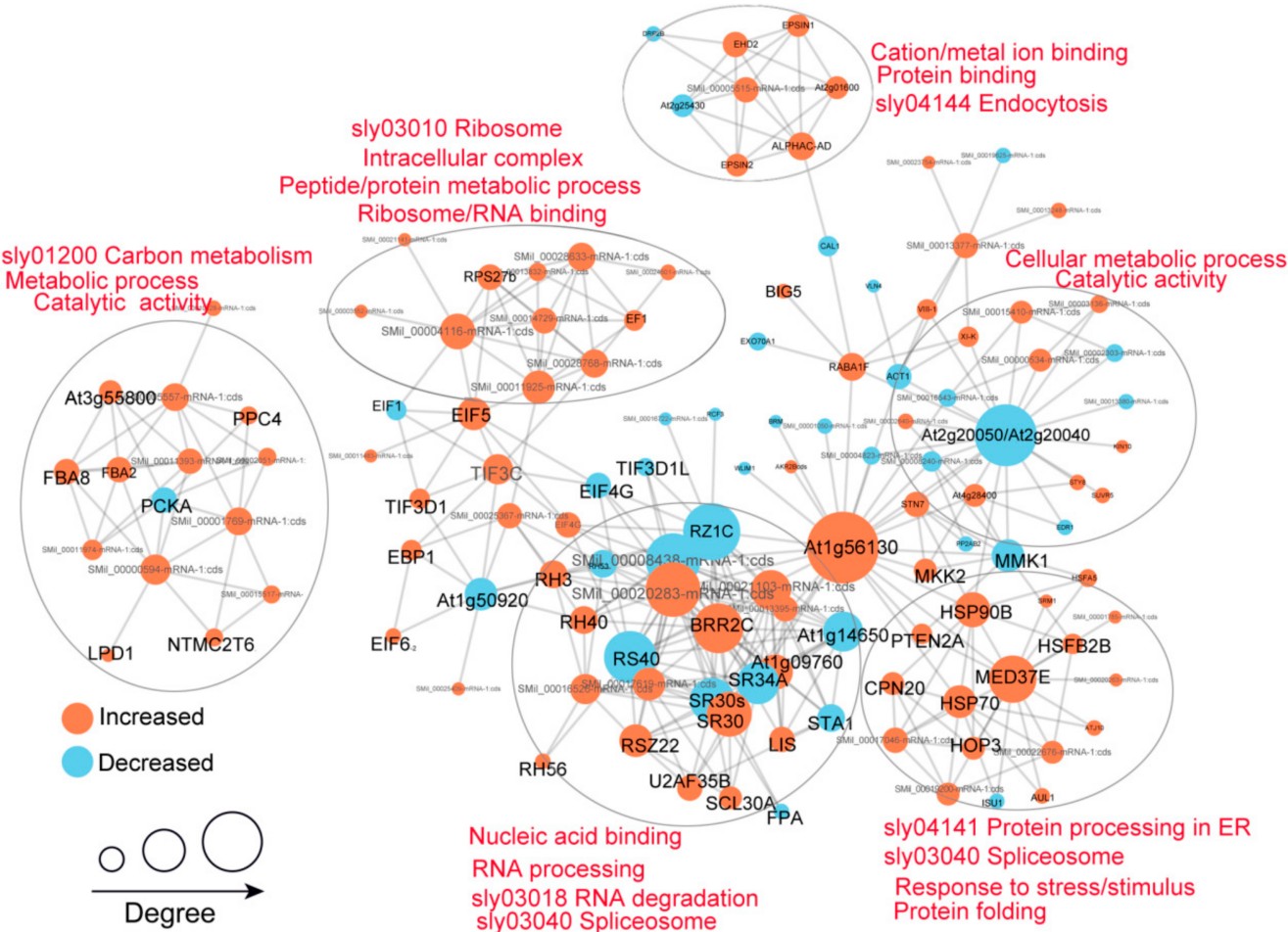

**Figure 5.** The protein-protein interaction network of significantly changed phosphoproteins in response to high drought stress. Blue and orange color notes phosphoproteins with significantly increased and decreased phosphorylation levels, respectively. Node size presents the interaction degree. The larger the node, the higher the degree of interaction is. Gray circles present the protein subsets with different functional classifications (red word). ER, endoplasmic reticulum.

The PPI network consisting of the 119 phosphoproteins common to HD and MD stresses contained 94 phosphoproteins (Figure S3) and three major subsets which were related to sly01200, sly03010, sly00196, sly03040, and sly04141.

*2.6. Phosphorylation Motifs of Significantly Changed Phosphoproteins in the PPI Network*

The phosphorylation-based motifs in the significantly changed phosphoproteins in the PPI network were identified and listed in Table 1. Motifs of [pSPxR], [pSxD], and [pSxG] were identified in 10, nine and eight proteins, respectively. The At1g56130 (a LRR receptor-like serine/threonine-protein kinase), HSP70, fructose-bisphosphate aldolase 8 (FBA8), and FAB2 proteins had increased phosphorylation levels in the [pSxG] motif. Ribosomal proteins and splicing factors had diverse and different phosphorylation motifs (Table 1). The RSZ22 protein had [pSPR] motif, and the SCL30A and U2AF35B proteins contained a [pSPxR] motif.

**Table 1.** Modified site feature sequence and enrichment statistics.

| Motifs | Protein Accession | Protein Description | Modified Sequences | Regulated (Stress/CK) |
|---|---|---|---|---|
| xxxxxx_S_xGxxxx | SMil_00024868 | Probable LRR receptor-like serine/threonine-protein kinase At1g56130 | LS(1)AGDLR | Up |
| | SMil_00002640 | Uncharacterized protein | VKQHT(1)LVS(1)GGVR | Up |
| | SMil_00004629 | Clathrin interactor EPSIN 2 | NPANEDDGQY(0.017)S(0.988)S(0.994)RGS(1)GAR | Up |
| | SMil_00006738 | Fructose-bisphosphate aldolase 8 (FBA8) | LS(0.035)S(0.965)INVENIEANR | Up |
| | SMil_00021644 | Fructose-bisphosphate aldolase 2 (FBA2) | LAS(1)IGLENTEANR | Up |
| | SMil_00009411 | Stromal 70 kDa heat shock-related protein (HSP70) | DAIS(0.968)GGS(0.031)T(0.001)QAMK | Up |
| | SMil_00013683 | Uncharacterized protein | S(1)GKELEK | Up |
| | SMil_00025367 | Uncharacterized protein | ITNPEPFLPLGS(0.006)DAFS(0.803)T(0.191)GK | Up |
| xxxxxx_S_xDxxxx | SMil_00003440 | Serine/threonine-protein kinase STN7 | NALAS(1)ALR | Up |
| | SMil_00023541 | Serine/threonine-protein kinase EDR1 | FVDS(0.987)LT(0.013)AEEK | Down |
| | SMil_00024588 | Phosphoenolpyruvate carboxylase 4 (PPC4) | ES(0.039)S(0.958)LDT(0.002)SK | Up |
| | SMil_00005515 | Uncharacterized protein | FDS(0.32)IS(0.686)S(0.984)S(0.01)R | Up |
| | SMil_00007312 | Brefeldin A-inhibited guanine nucleotide-exchange protein 5 (BIG5) | VT(0.006)GES(0.909)LS(0.085)NFEELK | Up |
| | SMil_00014509 | 40S ribosomal protein S27-2 (RPS27b) | LTEGCS(1)FR | Up |
| | SMil_00017266 | Uncharacterized protein | KLEDFEHSEAVAES(0.995)ENNS(0.005)DR | Down |
| | SMil_00019200 | Uncharacterized protein | ADTGDQS(1)EEK | Up |
| | SMil_00022967 | ATP-dependent helicase BRM | SGNGSAGPLS(0.98)PT(0.02)GIGR | Down |
| xxxxxx_S_DxExxx | SMil_00008933 | U4/U6 small nuclear ribonucleoprotein PRP4 (LIS) | T(0.044)AT(0.933)GHEY(0.023)EISEESK | Up |
| | SMil_00009506 | DEAD-box ATP-dependent RNA helicase 53 (RH53) | AGFAVADYS(1)DGER | Down |
| xxxxxx_S_PKxxxx | SMil_00006271 | Heat stress transcription factor A-5 (HSFA5) | LDEAQALS(0.017)S(0.907)S(0.076)PDANK | Up |

**Table 1.** *Cont.*

| Motifs | Protein Accession | Protein Description | Modified Sequences | Regulated (Stress/CK) |
|---|---|---|---|---|
| | SMil_00009223 | Eukaryotic translation initiation factor 4G (EIF4G) | S(0.01)LS(0.984)LES(0.006)PK | Up |
| | SMil_00011806 | EH domain-containing protein 2 (EHD2) | GTDAS(1)LPK | Up |
| xxxxxx_S_PRxxxx | SMil_00009056 | C2 domain-containing protein At1g53590 (NTMC2T6.1) | QIS(1)GQVR | Up |
| | SMil_00009156 | Serine/arginine-rich splicing factor RSZ22 | S(0.027)Y(0.014)S(0.907)RS(0.075)PY(0.978)R | Up |
| xxxxxx_S_PxRxxx | SMil_00024390 | Serine/arginine-rich splicing factor RS40 | KDGET(0.075)DHNHKHS(0.824)S(0.863)S(0.239)PQR | Down |
| | SMil_00005515 | Uncharacterized | FDS(0.32)IS(0.686)S(0.984)S(0.01)R | Up |
| | SMil_00011483 | Uncharacterized | HLS(0.807)T(0.193)DLPR | Up |
| | SMil_00013395 | Uncharacterized | LGFGLS(0.004)GS(0.996)GK | Up |
| | SMil_00013952 | Villin-4 (VLN4) | QNS(1)VDQSPK | Down |
| | SMil_00019792 | Hsp70-Hsp90 organizing protein 3 (HOP3) | EDTEMASNGS(1)PER | Up |
| | SMil_00023663 | Serine/arginine-rich SC35-like splicing factor SCL30A | APPPARS(1)PS(1)R | Up |
| | SMil_00023754 | Uncharacterized | HGS(1)VKPNEAAIPEGK | Up |
| | SMil_00030131 | Splicing factor U2af small subunit B (U2AF35B) | S(1)RS(1)PVREGS(1)EER | Up |
| | SMil_00022481 | DEAD-box ATP-dependent RNA helicase 40 (RH40) | S(0.007)FS(0.994)RS(1)PDR | Up |
| xxxxxx_S_xRxxxx | SMil_00028633 | 60S ribosomal protein L15 | NQTLS(1)LR | Up |
| | SMil_00023717 | Chaperone protein dnaJ 10 (ATJ10) | S(0.056)ANGS(0.753)NS(0.19)VPTDPPSHK | Up |
| xxxxxx_T_Pxxxxx | SMil_00011925 | 40S ribosomal protein S13 | T(0.999)PPS(0.001)WLK | Up |
| xxxxxx_S_FRxxxx | SMil_00028768 | 40S ribosomal protein S10-2 | GGAPADFQPS(1)FR | Up |
| xxxxRx_S_xxxxxx | SMil_00013683 | Elongation factor 1-alpha EF1 | S(1)GKELEK | Up |
| | SMil_00003660 | DEAD-box ATP-dependent RNA helicase 3 (RH3) | SPSFGGRDS(0.914)S(0.086)R | Up |
| xxxxxx_S_xxxRxx | SMil_00029002 | Histone-lysine N-methyltransferase SUVR5 | NIAS(1)PVNK | Up |

Table 1. *Cont.*

| Motifs | Protein Accession | Protein Description | Modified Sequences | Regulated (Stress/CK) |
|---|---|---|---|---|
| xxxxxx_S_Pxxxxx | SMil_00028660 | Exocyst complex component EXO70A1 | WDS(0.319)S(0.681)AS(1)EDAR | Up |
| xxxxxG_S_xxxxxx | SMil_00027479 | Serine/arginine-rich splicing factor SR34A | GS(1)GGGGGGGGGGSGSGGGR | Down |
| | SMil_00027680 | Probable mediator of RNA polymerase II transcription subunit 37e (MED3E) | MYQGAGGEVPMDDDAPAPS(0.144)GGS(0.856)GAGPK | Up |
| | SMil_00004116 | 30S ribosomal protein S9 | VVLQEGS(1)GK | Up |
| xxxxxx_T_Dxxxxx | SMil_00028092 | Protein STABILIZED1 (STA1) | GY(0.01)LT(0.99)DLK | Down |
| xxxxxx_S_xPxxxx | SMil_00001202 | Phosphoenolpyruvate carboxykinase (PCKA) | S(0.999)APT(0.59)T(0.411)PLNGSQGPFGAVSEDER | Down |

The numbers in parenthesis adjacent to the phosphorylation sites are phosphorylation probability.

### 2.7. Phosphorylation Profiles of Proteins Related to Terpenoid and Phenols Metabolism

Proteins related to terpenoid metabolism were selected from the literature. The phosphorylation levels of the phosphoproteins related to terpenoid metabolism are presented in Figure 6A. The phosphorylated levels of the Ser, Thr, or Tyr sites of proteins 1-deoxy-D-xylulose-5-phosphate synthase 2 (DXS2), isoprenylcysteine alpha-carbonyl methylesterase 1 (ICMEL1), and alpha-trehalose-phosphate synthase 8 (TPS8), Cytochrome P450 83B1 (CYP83B1) were reduced by drought stresses, and those of proteins TPS7, TPS11, 4-coumarate-CoA ligase-like 7 (4CLL7), Geraniol dehydrogenase 1 (GEDH1), and an uncharacterized protein (SMil_00019164) were increased by drought stress. The phosphorylation level of polyphenol oxidase (PPO) was increased by MD stress and that of (+)-larreatricin hydroxylase (LH) was decreased by HD and MD stresses (Figure 6A). Additionally, the phosphorylation levels of several members of transcription factor families MYB, bHLH, and MYC that were related to terpenoid and phenol metabolisms were changed upon drought stress (Figure 6B). These results indicated that the terpenoid metabolism in *S. miltiorrhiza* leaves might be regulated by drought stress.

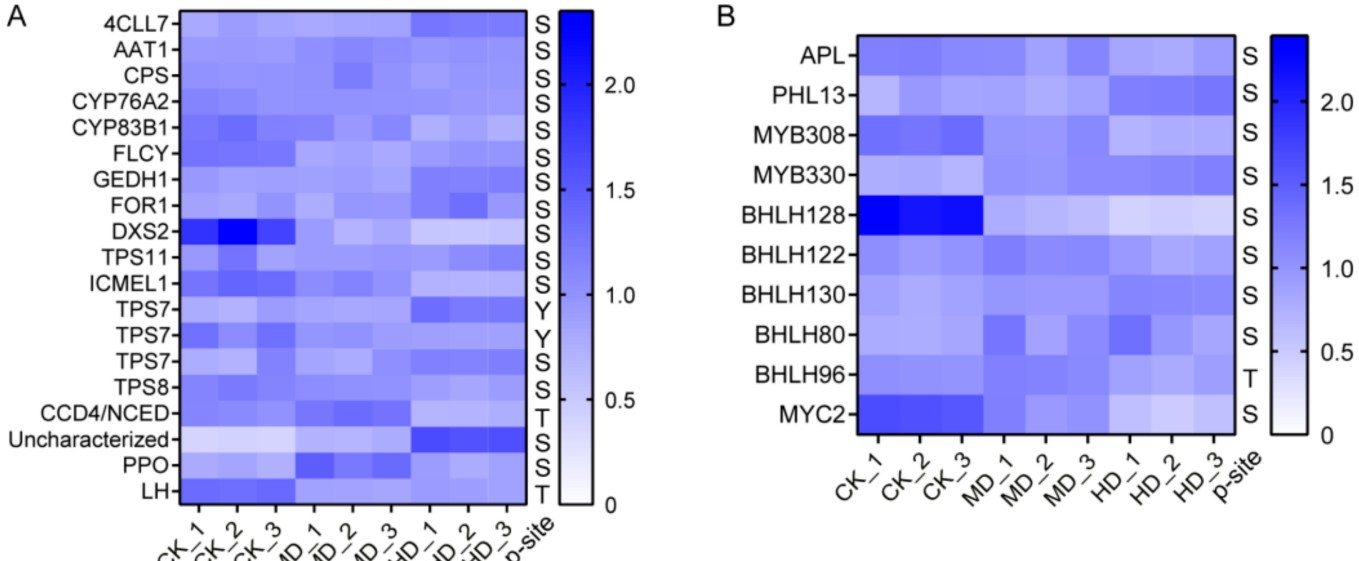

**Figure 6.** The phosphorylation profiles of significantly changed phosphoproteins in *Salvia miltiorrhiza* leaves. (**A**), the heatmap showing the changed phosphorylation profiles of proteins related to terpenoid and phenol metabolisms in *Salvia miltiorrhiza* leaves in response to drought stress (high drought, HD; and moderate drought, MD). (**B**), the heatmap showing the phosphorylation profiles of several transcription factors related to terpenoid and phenol metabolisms in *Salvia miltiorrhiza* leaves. Capital letters following heatmap indicate the phosphorylation sites (serine, S; tyrosine, Y; and threonine, T). CK, control.

### 2.8. Phosphorylation Profiles of Other Characteristic Proteins

At last, we identified the phosphorylation profiles of proteins related to energy metabolism, processing in the endoplasmic reticulum (ER), and RNA splicing (Figure 7A–C). The phosphorylation levels of most splicing factors, including RH56, RH3, SR30, SC35, SR34A, and RSZ22, were decreased or increased at the Ser or Tyr sites by drought stress (Figure 7A). For instance, the RSZ22 factor had increased Ser or Tyr phosphorylation levels upon drought stress (Figure 7A). Additionally, the phosphorylation levels of most metabolic proteins (Figure 7B) and HSPs (Figure 7C) had the highest phosphorylation level upon HD stress.

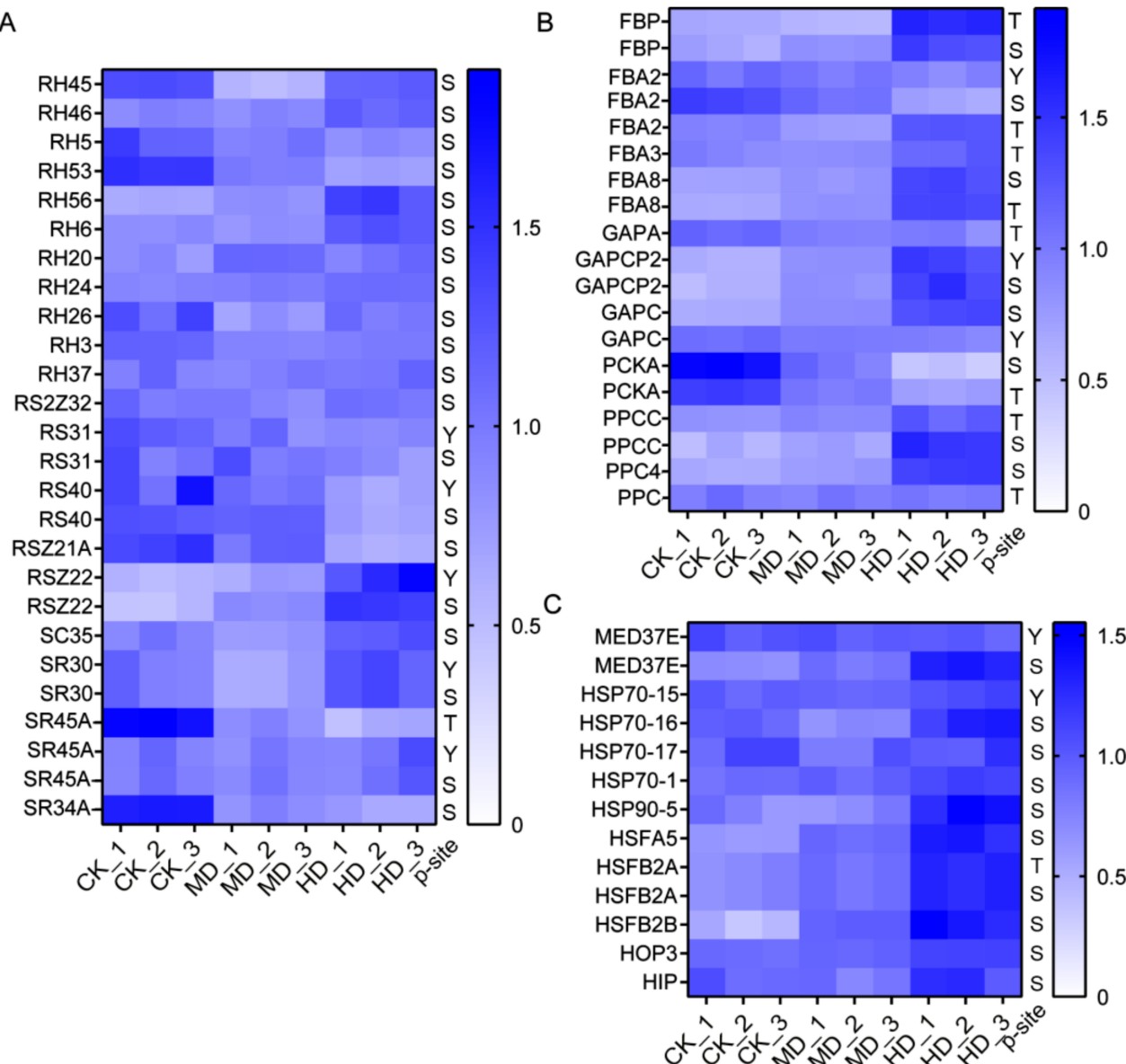

**Figure 7.** The phosphorylation profiles of proteins related to RNA splicing (**A**), energy metabolism (**B**), and heat shock proteins (HSPs, (**C**) in *Salvia miltiorrhiza* leaves. Capital letters following the heatmap indicate the phosphorylation sites (serine, S; tyrosine, Y; and threonine, T). HD, high drought, HD. MD, moderate drought. CK, control.

## 3. Discussion

*S. miltiorrhiza* is one of the most important and popular plants used in TCM. Research showed that drought stress significantly changed the growth of shoot and root, the shoot/root ratio, and the accumulation of active constituents in *S. miltiorrhiza* [22]. After withholding water for two weeks, the leaves of *S. miltiorrhiza* were slightly wilted and only older leaves were beginning to shrivel. After four weeks of withholding water, *S. miltiorrhiza* leaves were almost dried up, and the youngest leaves were beginning to shrivel. These results showed that drought treatment affected the normal growth of *S. miltiorrhiza*. To illustrate the physiological and phosphoproteomic changes in *S. miltiorrhiza* leaves upon drought stresses, LC-MS/MS proteomics technology was used in this study. It is expected to infer the molecular mechanism of *S. miltiorrhiza* in response to drought stress from the level of phosphoproteomics.

### 3.1. Phosphoproteins Involved in Carbon Metabolism and Energy Metabolism

Drought stress-induced inhibition in plant growth is mediated by various biological processes, including changes in morphology and molecular metabolism. Upon drought stress, plants can alleviate drought stress-induced injury by increasing the synthetic utilization of carbohydrates, which need to consume a lot of energy. A subset of the significantly changed phosphoproteins in *S. miltiorrhiza* leaves upon drought stress was enriched in carbon and carbohydrates metabolism, including PPC4, phosphoenolpyruvate carboxylase 2 (PPCC), FBA2, FBA8, and phosphoenolpyruvate carboxykinase (PCKA/PEPCK).

PCKA is highly expressed in response to cadmium stress [23,24]. PEPC catalyzes the irreversible conversion from β-carboxylation of phosphoenolpyruvate to oxaloacetate. PPC4 is involved in pyruvate metabolism. It converts pyruvate to oxaloacetate and promotes the synthesis of specific amino acids including glutamic acid and aspartic acid [25]. In addition, PPC4 is involved in the synthesis of malate. Both glutamic acid and malate were preferentially accumulated in plant cells under drought stress [3,26]. Gregory et al. [27] previously reported that phosphorylation activation of PPC1 in *A. thaliana* contributed to the metabolic adaptation to nutritional P(i) deprivation. We showed that high drought stress phosphorylated the PPC4 and PPCC proteins at the Ser and/or Thr site. However, the phosphorylation levels of PCKA Thr and Ser sites were decreased by drought stresses, especially the high drought stress. The changed phosphorylation levels of these proteins at different phosphorylation sites upon drought stress demonstrated that the energy metabolism in *S. miltiorrhiza* leaves was changed upon drought stress.

### 3.2. Phosphoproteins Involved in RNA Transport, Binding, and Splicing

Some studies have shown that the RNA-binding, chloroplast-transported DEAD-box RNA helicases (ATP-dependent), including RH53, RH58, RH3, and RH40 are involved in responses to many stresses and have the characteristics of multi-effects [28,29]. Existing studies have shown that the key of DEAD-box helicases in the regulation of cell stress resistance is that they can untwist the secondary structure of mRNA formed at 5′UTR and restore the linear structure of mRNA upon stresses, so as to promote their efficient translation [30,31]. Several studies also confirmed the roles of DEAD-box helicase genes in the splicing of Group II intron-containing RNAs from chloroplast in *Arabidopsis* [32–34]. The increased transcript levels of DEAD-box helicases by stresses contribute to the synthesis of intracellular membrane-bounded organelles which ultimately enhance resistance to drought, salt, low temperature, and heat stress [8,29,35]. Accordingly, these genes are candidate targets for the comprehensive improvement of crop stress resistance [8,36].

Studies confirmed the roles of DEAD-box helicase genes, including *AtRH3* in splicing of Group II intron-containing RNAs from chloroplast and large submit maturation of plastid ribosome in *Arabidopsis* [32–34]. For instance, the depletion of DEAD-box helicase genes impaired chloroplast and ribosome splicing [34]. Alternative splicing is an essential step for the function and maturation of mRNA from precursors by the spliceosome to remove non-coding intron sequences. The splicing complexes contain RNA small nuclear ribonucleoprotein complex (snRNP) and a variety of factors including pre-mRNA-processing factor, pre-mRNA-splicing factor, and splicing factors [37,38]. Splicing factors are involved in most stress and regulating the mechanism of environmental fitness and responses [39,40]. These factors affect the stability, transcription, and splicing of a subset of genes and plant rhythm [41,42]. In the present study, the phosphorylation levels of several serine/arginine-rich splicing factors including RSZ22, RS40, SCL30A, and SR34A were increased or decreased in *S. miltiorrhiza* leaves upon drought stress. These results demonstrated that drought stress changed the stability, transcription, and splicing of genes in *S. miltiorrhiza* leaves.

### 3.3. Phosphoproteins Involved in Processing in ER

HSPs are molecular chaperons composed of five families with a high affinity for unrolled or misfolded polypeptide chains. HSPs are responsible for protecting cells against

stress and conferring abiotic stress tolerance [43]. Their main function is to stabilize protein folding and prevent protein denaturation and aggregation caused by environmental stimuli [44,45]. HSPs associated with some molecules that are involved in signalings, like MAPK, Raf1, Akt, and calcium signaling [44]. In addition, HSP phosphorylation controls the expression of its target proteins and regulates the stress responses [46–48]. The phosphorylated HPS70 was reported to have decreased ATPase and refolding activity, and thus inhibited eukaryotic protein synthesis in yeast [49]. We confirmed that the phosphorylation levels of HSPs at the Ser residues were increased in *S. miltiorrhiza* leaves upon drought stress. The results of the increased Ser phosphorylation level of the HSPs indicated the upregulation of them in *S. miltiorrhiza* leaves in response to drought stress.

### 3.4. Phosphoproteins Associated with Terpenoid and Phenols Metabolisms

Enzymes related to terpenoid metabolism were identified from the phosphorylated proteins in *S. miltiorrhiza* leaves upon drought stress, including DXS2, Copal-8-ol diphosphate hydratase, chloroplastic (CPS), 4CLL7, GEDH1, and ICMEL1 [50,51]. Additionally, the members of transcription factor families which were related to the biosynthesis of terpenoids in *S. miltiorrhiza* leaves were identified, including bHLH, MYC, and MYB members [51–53]. Enzymes including TPS, DXS2, CPS, and 3-hydroxy-3-methylglutaryl-coenzyme A reductase (HMGR) were essential for the mevalonic acid (MVA) and 2-methyl-D-erythritol 4-phosphate (methylerythritol phosphate, MEP) pathways of terpenoid synthesis in plants [53–55]. For instance, TPSs act as metabolic gatekeepers in the evolution of terpenoid chemical diversity [56]. CPS controls the conversion from geranylgeranyl pyrophosphate to cobaki pyrophosphate and its inhibition suppresses the accumulation of tanshinones I and IIA [57]. The accumulation of terpenoids is regulated by the activities of these enzymes [54].

In addition, the overexpression of bHLH and MYB members, including *SmbHLH148*, *SmbHLH10*, *SmMYC2a*, *SmMYB9b*, and *SmMYB36*, enhances tanshinones or phenolic acids biosynthesis in *S. miltiorrhiza* [52,53,58–60]. Our present study identified that the Ser-phosphorylation levels of DXS2, MYC2, and bHLH128 were significantly decreased by drought stress. However, the Ser phosphorylation levels of 4CLL7, FOR1, GEDH1, MYB330, and bHLH130 proteins were slightly increased by drought stress. DXS2 is a key enzyme of the tanshinone biosynthesis pathway regulating the synthesis of tanshinones and terpenoids [61]. The decreased Ser-phosphorylation level of DXS2 might demonstrate the inhibition of tanshinone and terpenoid synthesis and accumulation in *S. miltiorrhiza* leaf.

In this present study, we demonstrated the different phosphorylation levels of proteins in *S. miltiorrhiza* leaves in response to drought stress. Among the phosphoproteins identified, the phosphorylation levels of proteins related to energy metabolism (including FBA2, FBA8, PCKA, and PPCC), RNA splicing (including RH5, RH6, and RSZ22), and terpenoid metabolism (including TPS7, DXS2, and bHLH transcription factors) were significantly changed in *S. miltiorrhiza* leaves upon drought stress, especially by high drought stress. Most of these proteins were phosphorylated at the Ser, Thr, and Tyr sites. This study highlighted the posttranscriptional mechanisms in *S. miltiorrhiza* leaves in response to drought stress.

## 4. Materials and Methods

### 4.1. Planting of S. miltiorrhiza and Drought Treatment

*S. miltiorrhiza* seeds were obtained from the experimental base of Shandong Agricultural University, Tai'an, China. The seeds were planted in plastic basins (12 cm × 10 cm) fulfilling matrix soil (dark loessial soil:sand:earthworm dung = 3:1:1) Experiments were performed in a laboratory under the cultivation conditions: $300 \ mol \cdot m^{-2} \cdot s^{-1}$ of illumination intensity, a 16:8 h light: dark cycle and a 65% relative humidity. After 20 days of seedling emergence, thinning was carried out and one seedling with a consistent growth state was selected from each pot. After 60 days of seedling emergence, *S. miltiorrhiza* plants with a consistent growth state were selected and subjected to drought treatment. The stimulation

of MD and HD stress was performed by simply withholding water for two and four weeks, respectively (Figure S1). Plants in the control group were treated with normal watering.

Each treatment was performed in triplicate with 10 plants of each replication. On the same day of the last watering, the plants of *S. miltiorrhiza* were photographed, and the young leaves were collected from three plants of each replication for each treatment/group. All the leaf samples were snapped in liquid nitrogen and then stored at −80 °C before extraction of protein.

### 4.2. Protein Extraction

Before protein extraction, an equal amount of leaf samples of each replication were pooled, and three samples for each treatment were prepared accordingly. Leaf samples were milled in liquid nitrogen and transferred into centrifuge tubes. Samples were diluted in lysis buffer (8 M urea, 1% Triton-100, 10 mM dithiothreitol, 1% protease inhibitor, 1% phosphatase inhibitors, and 2 mM EDTA; 1:4 $v/v$) and treated with sonication using a high-intensity ultrasonic processor (Scientz, Biotechnology, Ningbo, China) for three times. The remaining debris was removed by centrifugation at 20,000× $g$ at 4 °C for 10 min. Finally, the protein in the supernatant was precipitated in 20% of precooling trichloroacetic acid for 2 h at 4 °C and centrifuged at 12,000× $g$ 4 °C for 10 min. The supernatant was removed and protein precipitate was washed with precooling acetone three times. Protein samples were dissolved in urea (8 M) and the quantitative detection of protein concentration was performed using a bicinchoninic acid kit (Beyotime Biotechnology, Shanghai, China) according to the manufacturer's instructions.

### 4.3. Trypsin Digestion and Tandem Mass Tag (TMT) Labeling

Before digestion, the protein solution was incubated with dithiothreitol (5 mM; Sigma-Aldrich, Saint Louis, LA, USA) for 30 min at 56 °C, and alkylated with iodoacetamide (11 mM; Sigma-Aldrich) in dark for 15 min at room temperature. Then, tetraethylammonium bromide (TEAB, 100 mM; Sigma-Aldrich) was added to dilute protein, which was then subjected to two-step digestion (1: 50 ratio of trypsin to protein overnight and 1:100 ratio of trypsin to protein for 4 h). The desalination of the peptide was conducted using a Strata X C18 SPE column (Phenomenex, Torrance, CA, USA). After being vacuum-dried, the peptide was dissolved in TEAB (0.5 M; Sigma-Aldrich) and labeled using a TMT labeling kit (ThermoFisher Scientific, Waltham, USA) according to the manufacturer's protocol. Finally, peptides were fractionated into 60 fractions by high pH reverse-phase HPLC using Thermo Betasil C18 column (5 μm particles, 10 mm ID, and 250 mm length) with a gradient of 8%~32% acetonitrile over 60 min. Then, the peptides were combined into 6 fractions and vacuum-dried.

### 4.4. Affinity Enrichment

Affinity enrichment of peptide was performed as previously reported [62]. In brief, peptide solutions were incubated with immobilized metal affinity chromatography (IMAC) microspheres suspension with vibration in loading buffer (50% acetonitrile/6% trifluoroacetic acid) and enriched phosphopeptides were collected by centrifugation, and the supernatant was removed. The IMAC microspheres were washed with loading buffer (50% acetonitrile/6% trifluoroacetic acid and 30% acetonitrile/0.1% trifluoroacetic acid) to remove nonspecifically adsorbed peptides. Next, the enriched phosphopeptides were eluted from IMAC microspheres by vibration using elution buffer containing 10% NH$_4$OH. The supernatant containing phosphopeptides was collected and lyophilized for LC-MS/MS analysis.

### 4.5. LC-MS/MS Analysis

For LC-MS/MS analysis, peptides were dissolved in 0.1% formic acid (solvent A), separated by an EASY-nLC 1000 ultra-performance liquid chromatography (UPLC) system (Proxeon Biosystems, now Thermo Scientific) using a four-step linear gradient of 6% to

23% buffer B (98% acetonitrile and 0.1% formic acid; a flow rate of 350 nL/min) in 26 min, 23% to 35% in 8 min, and climbing to 80% in 3 min, and then holding at 80% for the last 3 min. A last, the separated peptides were subjected to MS/MS in Q Exactive^TM Plus (Thermo) coupled online to the UPLC. For MS analysis, the electrospray voltage was set at 2.0 kV. The $m/z$ scan range was 350 to 1800 for a full scan. The intact peptides were detected at a resolution of 70,000 at 100 $m/z$ and the fragments were detected at a resolution of 17,500 at 100 $m/z$. The duration of dynamic exclusion was 15.0 s Automatic gain control was set at $5 \times 10^4$.

### 4.6. Database Search

MS/MS spectra were processed using MaxQuant (v.1.5.2.8; http://www.maxquant.org/ accessed on 13 May 2019) on and searched against the National Data Center of Traditional Chinese Medicine of China (http://www.ndctcm.org/ accessed on 13 May 2019). Protein identification was conducted following the criteria: precursor ion mass tolerances = 20 ppm in first search and 5 ppm in main search; mass tolerance for fragment ions = 0.05 Da; enzyme = trypsin; missing cleavage = 4; fixed modification: carbamidomethyl (C) on Cys; variable modification: oxidation (M), phosphorylation (S/T/Y). The false discovery rate threshold for peptide and protein identification was set at <1%. The minimum peptide length was set to 40. Phosphosites with a localization probability of larger than 0.75 were retained and used for later analysis.

### 4.7. Bioinformatics Analysis

Significant changes in phosphorylation levels were identified with the criteria of fold change >1.5 or <1/1.5 and $p < 0.05$ (two-tailed *T*-test). The distinct expression of proteins with significant changes in phosphorylation levels was presented using heatmap clustering by R Package pheatmap (v.2.0.3; https://cran.r-project.org/web/packages/cluster/ accessed on 13 May 2019). The phosphorylation motifs and its specificity were extracted using the Motif-X algorithm in MoMo (V5.0.2; http://meme-suite.org/tools/momo accessed on 13 May 2019) with the criteria of peptide length = 21, occurrence = 20, and *p*-value < 0.000001. The Gene Ontology functional enrichment of the identified proteins and protein domain annotation was performed using InterProScan (v.5.14-53.0; http://www.ebi.ac.uk/interpro/ accessed on 13 May 2019). KEGG pathways that related to proteins were identified using KAAS (v.2.0; http://www.genome.jp/kaas-bin/kaas_main accessed on 13 May 2019) and KEGG Mapper (v2.5; http://www.kegg.jp/kegg/mapper.html accessed on 13 May 2019). The subcellular localization was predicted using WolfPsort (http://wolfpsort.org accessed on 13 May 2019). Gene Ontology biological processes and KEGG pathways that relate to the proteins with significant changes in phosphorylation levels were predicted with the threshold of $p < 0.05$. The PPI networks were identified in the online available program STRING (http://string-db.org/ accessed on 13 May 2019). Interactions with confidence score > 0.7 (high confidence) were retained and used for the construction of the PPI network using the networkD3 of R package (v.0.4 https://cran.r-project.org/web/packages/networkD3/ accessed on 13 May 2019). The protein names were presented by the Protein accession number in STRING, or protein names or locus numbers in *Arabidopsis thaliana* (http://www.arabidopsis.org accessed on 13 May 2019).

**Supplementary Materials:** The following supporting information can be downloaded at: https://www.mdpi.com/article/10.3390/agronomy12040781/s1, Figure S1. The schematic diagram of the experimental design and process in this study. * notes the date of sampling. MD and HD indicate moderate and high drought stress, respectively. CK, control. Figure S2. The distinct expression profiles of the proteins had significant changes in the phosphorylation level upon drought stresses. MD and HD indicate moderate and high drought stress, respectively. CK, control. Figure S3. The protein-protein interaction network of significantly changed phosphoproteins upon high versus moderate drought stress. Blue and orange color notes phosphoproteins with significantly increased and decreased phosphorylation levels, respectively. Node size presents the interaction degree. The

larger the node, the higher the degree of interaction is. The three gray circles present the protein subsets with different functional classifications (red word).

**Author Contributions:** Conceptualization: J.Z. and J.W.; Methodology: J.Z. and J.L.; Software and formal Analysis: J.Z., Y.S. and J.L.; Investigation: Y.S. and Z.S.; Resources and data curation: J.Z. and J.L.; Writing-original draft preparation: J.Z.; Writing—review & editing: Z.S., Y.S. and J.W.; Funding acquisition: Z.S. and J.W.; Supervision: J.W. Project administration: J.Z. All authors have read and agreed to the published version of the manuscript.

**Funding:** This work was supported by the National Natural Science Foundation of China (81872949), the Natural Science Foundation of Shandong Province of China (ZR2019HM081), the Key Research and Development Plan of Shandong Province (2019LYXZ021; 2017CXGC1302), Shandong modern agricultural industry technical system project (SDAIT-20-04), and National Key R&D Program of China (2017YFC1702705).

**Data Availability Statement:** Raw data is available from the corresponding author.

**Conflicts of Interest:** The authors declare no conflict of interest.

## Abbreviations

| | |
|---|---|
| 4CLL7 | 4-coumarate-CoA ligase-like 7 |
| DXS2 | 1-deoxy-D-xylulose-5-phosphate synthase 2 |
| FBA | fructose-bisphosphate aldolase |
| FC | fold change |
| FDR | false discovery rate |
| GEDH1 | Geraniol dehydrogenase 1 |
| HD | high drought |
| HMGR | 3-hydroxy-3-methylglutaryl-coenzyme A reductase |
| HSPs | heat shock proteins |
| ICMEL1 | isoprenylcysteine alpha-carbonyl methylesterase 1 |
| IMAC | immobilized metal affinity chromatography |
| KEGG | Kyoto Encyclopedia of Genes and Genomes |
| LC-MS/MS | liquid chromatography–tandem mass spectrometry |
| MD | moderate drought |
| MEP | 2-methyl-D-erythritol 4-phosphate |
| MS/MS | tandem mass spectrometry |
| MVA | mevalonic acid |
| PCKA/PEPCK | phosphoenolpyruvate carboxykinase |
| PPCC | phosphoenolpyruvate carboxylase 2 |
| PPI | protein-protein interaction |
| TPS | alpha-trehalose-phosphate synthase |

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
