# Peer review of "The Posttranscriptional Mechanism in Salvia miltiorrhiza Bunge Leaves in Response to Drought Stress Using Phosphoproteomics"

_agronomy, doi:10.3390/agronomy12040781_

Round 1
Reviewer 1 Report
Line 14: italicize "S. miltiorrhiza".
Line 15: give the full meaning of "TMT"; first time usage.
Line 68-69: The sentence is incomplete.
Page 12, line 96: Change "increased or increased" to "increased or decreased"
Page 12, line 99: Write the full meaning of "ER"
The paper is well written!
Author Response
Line 14: italicize "S. miltiorrhiza".
Response: "S. miltiorrhiza" here and other places are checked and italicized.
Line 15: give the full meaning of "TMT"; first time usage.
Response: The full name tandem mass tags had been added in Abstract and Methods.
Line 68-69: The sentence is incomplete.
Response: This sentence has been recast, by deleting the redundant word ‘how’.
Page 12, line 96: Change "increased or increased" to "increased or decreased"
Response: This correction has been done as suggested.
Page 12, line 99: Write the full meaning of "ER"
Response: ER is for endoplasmic reticulum.
Reviewer 2 Report
The Manuscript entitled “The posttranscriptional mechanism in Salvia miltiorrhiza Bunge leaves in response to drought stress using phosphoproteomics” by Zhang et al., reported the phosphoproteomic changes in Salvia miltiorrhiza upon drought treat.
Salvia miltiorrhiza is an important medicine plant. Quantitative phosphoproteomics studies will help reveal molecular mechanism in drought response and tolerance. The authors proposed the possible function of photosynthesis, RNA splicing and terpene metabolism in adaption to drought stress. The manuscript was well-written. I have a few comments below.
- In Figure 1, phenotypic change is not sufficient to describe the intensity of drought stress. Physiological data, such as water loss, is encouraged to be included.
- The method on phosphopeptide enrichment is too simple.
- In Figures 6 and 7, “CKY”, “MDY”, and “HDY” should be denoted in the legends.
- Line 150, a full-stop is missing.
Author Response
The Manuscript entitled “The posttranscriptional mechanism in Salvia miltiorrhiza Bunge leaves in response to drought stress using phosphoproteomics” by Zhang et al., reported the phosphoproteomic changes in Salvia miltiorrhiza upon drought treat.
Salvia miltiorrhiza is an important medicine plant. Quantitative phosphoproteomics studies will help reveal molecular mechanism in drought response and tolerance. The authors proposed the possible function of photosynthesis, RNA splicing and terpene metabolism in adaption to drought stress. The manuscript was well-written. I have a few comments below.
- In Figure 1, phenotypic change is not sufficient to describe the intensity of drought stress. Physiological data, such as water loss, is encouraged to be included.
Response: Sorry. We did not detect the changes in Physiological parameters during stress. The lack of Physiological data is a shortcoming of this paper.
- The method on phosphopeptide enrichment is too simple.
- Response: This section has been modified and more information has been added.
- In Figures 6 and 7, “CKY”, “MDY”, and “HDY” should be denoted in the legends
- Response: The abbreviations have been changed to CK (control), MD (moderate drought), and HD (high drought).
- Line 150, a full-stop is missing.
Response: This sentence has been recast: “The HD stress upregulated the phosphorylation levels of proteins related to sly00710, sly04141, sly00196, and “Carbon metabolism” compared with MD stress (sly01200; Figure 4B)”.